# Effect of tailoring anticoagulant treatment duration by applying a recurrence risk prediction model in patients with venous thromboembolism compared to usual care: A randomized controlled trial

**Geert-Jan Geersing**[1]*, **Janneke M. T. Hendriksen**[1], **Nicolaas P. A. Zuithoff**[1], **Kit C. Roes**[1,2], **Ruud Oudega**[1], **Toshihiko Takada**[1], **Roger E. G. Schutgens**[3], **Karel G. M. Moons**[1]

1 Julius Center for Health Sciences and Primary Care, University Medical Center Utrecht, Utrecht University, Utrecht, the Netherlands, 2 Biostatistics Research Group, Department of Health Evidence, Radboud University Medical Center, Radboud University, Nijmegen, the Netherlands, 3 Van Creveld Clinic, University Medical Center Utrecht, Utrecht University, Utrecht, the Netherlands

* g.j.geersing@umcutrecht.nl

**Data Availability Statement:** Data from this randomized controlled trial cannot be shared

## Abstract

### Background

Patients with unprovoked (i.e., without the presence of apparent transient risk factors such as recent surgery) venous thromboembolism (VTE) are at risk of recurrence if anticoagulants are stopped after 3–6 months, yet their risk remains heterogeneous. Thus, prolonging anticoagulant treatment should be considered in high-risk patients, whereas stopping is likely preferred in those with a low predicted risk. The Vienna Prediction Model (VPM) could aid clinicians in estimating this risk, yet its clinical effects and external validity are currently unknown. The aim of this study was to investigate the clinical impact of this model on reducing recurrence risk in patients with unprovoked VTE, compared to usual care.

### Methods and findings

In a randomized controlled trial, the decision to prolong or stop anticoagulant treatment was guided by predicted recurrence risk using the VPM (*n* = 441), which was compared with usual care (*n* = 442). Patients with unprovoked VTE were recruited from local thrombosis services in the Netherlands (in Utrecht, Harderwijk, Ede, Amersfoort, Zwolle, Hilversum, Rotterdam, Deventer, and Enschede) between 22 July 2011 and 30 November 2015, with 24-month follow-up complete for all patients by early 2018. The primary outcome was recurrent VTE during 24 months of follow-up. Secondary outcomes included major bleeding and clinically relevant non-major (CRNM) bleeding. In the total study population of 883 patients, mean age was 55 years, and 507 (57.4%) were men. A total of 96 recurrent VTE events (10.9%) were observed, 46 in the intervention arm and 50 in the control arm (risk ratio 0.92, 95% CI 0.63–1.35, *p* = 0.67). Major bleeding occurred in 4 patients, 2 in each treatment arm,

publicly because this study includes human research participant data and prior to sharing data, consulting an ethics committee is needed to ensure data are shared in accordance with participant consent and all applicable local laws. Data are available from the Institutional Data Access (contact via SecretariaatHAG-Onderzoek@umcutrecht.nl) for researchers who meet the criteria for access to confidential data, accompanied with a protocol describing the research questions aimed to answer.

**Funding:** The study received funding from the Netherlands Organisation for Health Research and Development (ZonMw grant number 171002214, KGMM was the receiving grant holder, URL: www.zonmw.nl). KGMM received a grant from The Netherlands Organization for Scientific Research (ZonMw 91810615 and 91208004; URL: www.zonmw.nl). GJG is supported by a VENI and VIDI grant from the Netherlands Organisation for Health Research and Development (ZonMw numbers 016.166.030 and 016.196.304; URL: www.zonmw.nl). The funders had no role in study design, data collection and analysis, decision to publish, or preparation of the manuscript.

**Competing interests:** The authors have declared that no competing interests exist.

**Abbreviations:** CRNM, clinically relevant non-major; DVT, deep vein thrombosis; ISTH, International Society on Thrombosis and Haemostasis; ITT, intention to treat; NOAC, non-vitamin K antagonist oral anticoagulant; PE, pulmonary embolism; VPM, Vienna Prediction Model; VTE, venous thromboembolism.

whereas CRNM bleeding occurred in 20 patients (12 in intervention arm versus 8 in control arm). The VPM showed good discriminative power (c-statistic 0.76, 95% CI 0.69–0.83) and moderate to good calibration, notably at the lower spectrum of predicted risk. For instance, in 284 patients with a predicted risk of >2% to 4%, the observed rate of recurrence was 2.5% (95% CI 0.7% to 4.3%). The main limitation of this study is that it did not enroll the pre-planned number of 750 patients in each study arm due to declining recruitment rate.

## Conclusions

Our results show that application of the VPM in all patients with unprovoked VTE is unlikely to reduce overall recurrence risk. Yet, in those with a low predicted risk of recurrence, the observed rate was also low, suggesting that it might be safe to stop anticoagulant treatment in these patients.

## Trial registration

Netherlands Trial Register NTR2680

## Author summary

### Why was this study done?

- Patients with clots in the leg or lung without a clear explanation are treated with anticoagulants.

- After the initial treatment phase is complete (after 3–6 months), a relatively large group of patients will then experience new clots if treatment is then discontinued.

- Physicians struggle with identifying the group of patients in need of prolonged anticoagulant treatment, which can be protective for recurrent clots, yet also inherently introduces bleeding risk.

### What did the researchers do and find?

- The researchers performed a randomized trial comparing the use of a prediction tool with usual care.

- This prediction tool can help identify patients at increased risk of recurrent clots. Treatment can then be tailored accordingly.

- The study found that, overall, the risk of recurrent clots was not reduced by applying this tool. The prediction tool, however, was able to identify a group of patients with a low risk of recurrent clots.

### What do these findings mean?

- On a population level, the researchers were unable to find a difference in the risk of recurrent clots by using this prediction tool.

- The study, however, did suggest that it might be safe to stop anticoagulant treatment after the initial treatment phase in those with a low predicted risk of recurrent clots.

## Introduction

Venous thromboembolism (VTE)—i.e., deep vein thrombosis (DVT) or pulmonary embolism (PE)—is a major healthcare burden [1]. An initial course of treatment for 3 to 6 months is recommended to prevent clot progression and VTE-related acute morbidity and mortality. After this initial phase, prolonging anticoagulant treatment reduces recurrence risk but at the price of increasing bleeding risk. Notably, patients with unprovoked VTE may benefit from prolonging treatment as it is widely appreciated that recurrence risk in them is highest [2]. However, both recurrent VTE events and major bleeding—reduced and induced by prolonged anticoagulation, respectively—carry a substantial mortality and morbidity risk. Clinically it is often difficult to balance both risks optimally on an individual level.

A Cochrane review with meta-analysis reported that currently there is insufficient evidence to support prolonging anticoagulant treatment in all patients with unprovoked VTE [3]. The main reason for this is that VTE recurrence risk in the mixed population of patients with unprovoked VTE is too heterogeneous. Ideally, therapeutic decision-making in these patients should be guided by individualized risks, whereby prolonging treatment is only considered in those with a high expected recurrence risk.

We therefore designed the VISTA study. The primary aim was to evaluate the effect of a risk-tailored patient management approach, compared to usual care, in an open-label, pragmatic randomized trial. The risk-based treatment approach included the use of the previously developed Vienna Prediction Model (VPM). This prognostic model predicts VTE recurrence risk in patients with unprovoked VTE [4], and based on this predicted risk, anticoagulant treatment was only continued if the predicted annualized risk was equal to or higher than 5%. This index strategy was compared with usual care as control strategy, which at the time of VISTA enrollment consisted of stopping anticoagulant treatment after the initial treatment phase in most patients, yet left this decision at the discretion of the treating physician. A secondary aim was to externally validate the VPM in a different setting and country from where it was derived (i.e., Austria) for its ability to predict VTE recurrence in terms of discrimination and calibration.

## Methods

### Trial design and ethical approval

VISTA was an investigator-initiated, multi-center, pragmatic, unblinded randomized trial. Patients were recruited by local thrombosis services in the Netherlands (in Utrecht, Harderwijk, Ede, Amersfoort, Zwolle, Hilversum, Rotterdam, Deventer, and Enschede). The trial was designed and overseen centrally at the University Medical Center Utrecht, the Netherlands. The study protocol was approved by the medical ethics research committee from the

University Medical Center Utrecht, and the trial was registered in the Netherlands Trial Register (https://www.trialregister.nl) with registration number NTR2680. The protocol is available in S1 Text. The trial was performed in accordance with the principles of the Declaration of Helsinki. All participants provided written informed consent. Throughout this paper, we adhere to the CONSORT reporting guidelines for randomized controlled clinical trials (see S1 CONSORT Checklist) [5].

## Participants

Adult patients (age 18 years or higher) with unprovoked VTE were eligible for participation in this trial. Patients were recruited by 9 thrombosis services throughout the Netherlands (in Utrecht, Harderwijk, Ede, Amersfoort, Zwolle, Hilversum, Rotterdam, Deventer, and Enschede) between 22 July 2011 and 30 November 2015. Unprovoked was defined as the patient having had no previous surgery (lasting longer than 30 minutes under general or spinal anesthesia) or casting after lower-limb injury within 3 months prior to the VTE event, and not being pregnant or in puerperium (up to 6 weeks postpartum). Oral-contraceptive-related VTE in female patients was allowed, given the uncertainty whether these events should be classified as provoked or unprovoked [6,7]. Patients with a history of VTE within the past 10 years were excluded, as were patients with an active malignancy (i.e., latest treatment less than 6 months prior to the VTE event or in palliative care): In both of these patient groups, there is not much debate that recurrence risk is high, warranting prolonged anticoagulant treatment. Finally, patients were also excluded from participation in this trial if there was a clear other indication for anticoagulant treatment at the time of the VTE event, such as atrial fibrillation or prosthetic heart valves.

## Randomization and blinding

All participants included in the study first followed an initial treatment course with vitamin K antagonists for 3 to 6 months, after low-molecular-weight heparin (LMWH) lead-in. Shortly before the end of this initial treatment phase—i.e., after 3–6 months for most patients—participants were randomized 1:1 to the index strategy (risk-tailored approach) or the control strategy (usual care). Patients were deliberately randomized at the end of the initial treatment phase because in some patients new clinical circumstances—such as diagnosis of an occult malignancy during the first treatment months—may dictate whether or not prolonged treatment is potentially recommendable. Blocked randomization was performed within strata for center and for type of VTE. Blinding of the participants and treating physicians was not possible given the pragmatic nature of this study and the index strategy.

## Index group

In the risk-tailored management strategy, the VPM was used to estimate the risk (or probability) of recurrence. The 3 predictors of the model were measured from each patient randomized to the index strategy and were used to estimate the patient's recurrence risk. These predictors are the extent of VTE (defined as distal DVT, proximal DVT, or PE), patient sex, and D-dimer concentration (on a continuous scale measured as micrograms/liter). Results from this risk prediction were communicated over phone to the patient, followed by a discussion on the clinical consequences of this risk. Subsequently, the treating physician was informed, along with being given an explanation of the risk model used. These phone conversations were performed by the 2 medical doctors (GJG or JMTH) with knowledge of prediction research and thrombosis management. For an illustration of how the model was applied to individual patients, see http://www.meduniwien.ac.at/user/georg.heinze/zipfile/ViennaPredictionModel.html.

Based upon consensus from the Scientific and Standardization Committee (SSC) of the International Society on Thrombosis and Haemostasis (ISTH), extending treatment in the index group was recommended only if the annualized predicted risk of recurrence was equal to or higher than 5%. Otherwise, treatment was discontinued [8]. We explicitly opted for a so-called model-assisted treatment recommendation (in which both the patient and treating physician discussed the predicted risk to determine whether or not to continue in shared decision-making), instead of a directive or enforced recommendation [9].

Risk estimation was performed twice, namely shortly before treatment cessation at the end of the initial treatment period (thus while still using anticoagulant treatment; VPM 1) and approximately 1 month after treatment cessation (VPM 2). This second assessment was done because D-dimer levels can be lowered by the use of anticoagulant treatment, thus falsely leading to lower predicted risk of VTE recurrence (higher D-dimer levels correlate with higher risk). This second assessment was skipped if the first risk estimation already identified a patient as high risk.

## Control group

In patients randomized to the control group, usual care was followed. This meant that the decision to stop or prolong anticoagulant treatment after the initial treatment phase was left at the discretion of the treating physician and patient, conforming to prevailing current care. This was thus without a formally guided risk estimation and communication.

## Outcome measures

The primary outcome was the occurrence of recurrent VTE during 24 months of follow-up after randomization, thus during the 24 months after the completion of the initial anticoagulant treatment phase. Recurrent VTE was defined as proximal DVT or fatal or nonfatal PE, as confirmed by compression ultrasonography for DVT and by computer tomographic pulmonary angiography (CTPA) for PE, accompanied by management with anticoagulation treatment. Outcome assessment was done after scrutinizing all available hospital discharge information. Given the pragmatic nature of our trial, outcomes were not adjudicated blinded to randomized allocation.

Secondary outcomes were the occurrence of major bleeding and clinically relevant nonmajor (CRNM) bleeding, as reported by the patient or treating physician. Prompted by the reporting of bleeding by either the patient or his/her physician, bleeding was classified as major if it was retroperitoneal or intracranial bleeding, was accompanied by a lowering of hemoglobin level of at least 20 g/l, or resulted in transfusion of at least 2 units of blood, surgical intervention, or invasive procedures to stop the bleeding. Data on CRNM bleeding (all bleeding events not meeting the criteria stated above for which a physician consultation was needed, e.g., epistaxis, large hematoma, or rectal bleeding) were collected by questionnaires sent to patients by mail.

## Follow-up appointments

Patients were followed up by phone at 3, 12, and 24 months after randomization, thus for 24 months in total after the completion of the initial treatment phase of (typically) 3–6 months. They were also instructed to call a research nurse in case of a potential recurrent VTE event, bleeding event, or any other serious adverse event. When we were unable to reach patients over the phone at one of the follow-up time points, we contacted their primary care provider to collect clinical information on the occurrence of recurrent VTE and bleeding. If a recurrent event was suspected, i.e., either spontaneously reported by the patient or detected during one

of the scheduled contacts, all additional clinical data were retrieved including information from general practitioners and hospital discharge information. Next, if such medical information clearly indicated recurrent DVT or recurrent PE (using our above-described definitions), a patient was classified as having had a recurrent event, after which follow-up was discontinued.

## Statistical analyses and power calculations

In a randomized trial on D-dimer-guided duration of anticoagulant treatment, patients with a persisting (over 1 month) positive D-dimer test after the initial treatment phase (i.e., "high risk patients") were randomized to prolonged (intervention) or discontinued anticoagulant treatment (control) [10]. In this study, the observed occurrences (during a median follow-up of 18 months) of recurrent VTE in the index versus control group were 15.0% versus 2.9%, with a risk reduction of more than 80% with the intervention [10]. We used a different design in which we randomized patients to whether or not a risk-based strategy was applied, thereby also including patients with a low predicted risk. Accordingly, we assumed a lower risk reduction than 80%, namely 50%. We conservatively estimated the occurrence of recurrent VTE in our control group would be 7% during follow-up. Thus, at a 2-sided alpha of 0.05 and a power of 80%, 692 patients in each treatment arm were needed given our assumed relative risk reduction of 50%, leading to a target population of 750 patients per treatment arm allowing for loss to follow-up.

The primary analysis (following the intention to treat [ITT] principle) compared the incidence of VTE recurrence within 24 months after the initial anticoagulant therapy phase—in both patient groups, presented as a risk ratio, as well as a risk difference (both with corresponding 95% confidence intervals). Both the risk ratio and the risk difference were estimated with generalized linear models for binomial distributions with the log link and identity link, respectively. Next, given that we had an assistive risk-based index approach, we also wanted to explore non-adherence to the recommended treatment strategy. This secondary analysis was not preplanned in the protocol, but was deemed informative by the study group to explore treatment effectiveness in those more likely to be adherent to the intervention strategy. We first fitted a propensity score model (in the intervention group only) with protocol non-adherence as binary outcome, defined as either not obtaining a D-dimer measurement or not following the treatment recommendation from the VPM. We applied logistic regression analysis, using the prognostic factors from the VPM as predictors (for D-dimer concentration using natural cubic splines as the association between D-dimer concentration and non-adherence was not linear), in addition to oral contraceptive use given the imbalance between groups (see S1 Table). The resulting propensity score model was then applied in all patients to estimate the probability of protocol non-adherence in both arms of the trial for each imputed dataset separately [11]. Next, the primary ITT analysis was repeated, comparing patients in the intervention and control arms with a similar estimated high probability of protocol adherence, i.e., patients within the 75th quartile of predicted probability of protocol adherence, thus keeping the randomization status in place.

Additionally, we evaluated the predictive performance of the VPM. For this analysis, we used only the data of patients in the index and control groups who did not prolong anticoagulant treatment, to account for the treatment effect of anticoagulation on the incidence of the outcome. Discrimination and calibration were assessed using the c-statistic and a calibration plot for the outcome at 12 months of follow-up, using the baseline hazard function of the VPM at this time point, which we received from the authors of the original publication. This was done deliberately as thereby we explicitly used the same baseline hazard and prediction model

that was used in the original derivation paper as well as in the index arm of our trial, as there also the predicted risk of recurrent VTE at 12 months ($\geq$5% versus <5%) was used to recommend decisions on prolonged treatment. Finally, post hoc and prompted by the reviewers from *PLOS Medicine* and using the full spectrum of 24 months of trial follow-up, we plotted the observed risk of recurrent VTE in a Kaplan–Meier graph for different predicted risks of recurrent VTE based upon the VPM (i.e., for an annualized predicted risk of 0% to 2%, >2% to 4%, >4% to 5%, and above 5%). These additional analyses are described in more detail in S2 Text.

## Missing values

Both for the explorative analysis of the impact of non-adherence to the recommended strategy based upon the VPM (fitting of the propensity model, see above) and for the validation analysis of the VPM, missing values for D-dimer concentration were imputed using chained equations. D-dimer concentration was missing in 251 patients (28.4%), and these missing values were imputed using all available information including the patients' baseline characteristics as well as the observed outcome. Ten imputed sets were created, and the results of analyses were pooled based on methodological recommendations.

## Results

In total 957 patients were deemed potentially eligible for inclusion. After an initial treatment phase, which lasted 3 to 6 months in the vast majority of patients, 883 patients were randomized to the intervention ($n$ = 441) or control ($n$ = 442) (Fig 1). The study did not enroll the preplanned number of 750 patients per treatment arm. Declining patient accrual into the study, mainly due to the changing anticoagulant landscape with the introduction of non-vitamin K antagonist oral anticoagulants (NOACs), which are not managed by thrombosis services in the Netherlands, in addition to budget constraints, prevented us from continuing inclusion into the trial. In close collaboration with our data and safety monitoring board, as well as the granting organization, we therefore decided to end inclusion on 30 November 2015. Thus, given that patients were treated first for 3–6 months, after which they were randomized and followed up for 24 months, follow-up of all study participants was complete in early 2018.

Mean age of the included 883 patients was 55 years, and in total 443 patients (218 in the index group and 216 in the control group) had proximal or distal DVT as the index event, and 440 patients had PE with or without DVT. Patient characteristics were well balanced between the groups (Table 1).

## Outcomes and effect measures

During 24 months of follow-up, 96 recurrent proximal DVT or PE events were observed (10.9%): 46 in the intervention arm (10.4%) and 50 in the control arm (11.3%), yielding a risk ratio of 0.92 (95% CI 0.63 to 1.35, $p$ = 0.67) and risk difference of −0.8% (95% CI −5.0% to 3.2%).

Major bleeding (after randomization) occurred in 4 patients (no fatal events): 2 major bleeding events in the intervention arm (i.e., 0.5%, 95% CI 0.1%–1.6%) and 2 in the control arm (i.e., 0.5%, 95% CI 0.1%–1.6%). CRNM bleeding occurred in 20 patients, 12 in the intervention arm (i.e., 2.7%, 95% CI 1.4%–4.7%) and 8 in the control arm (i.e., 1.8%, 95% CI 0.8%–3.5%), during any time point of follow-up. Based upon the VPM risk estimation, a total of 124 patients were labeled as having a high risk of recurrence and followed the VPM recommendation to continue anticoagulant treatment. During follow-up, this number gradually dropped to 111 patients still using anticoagulant treatment at the end of follow-up. In the control group a smaller number of 65 patients, declining to 35 patients at the end of follow-up, also continued

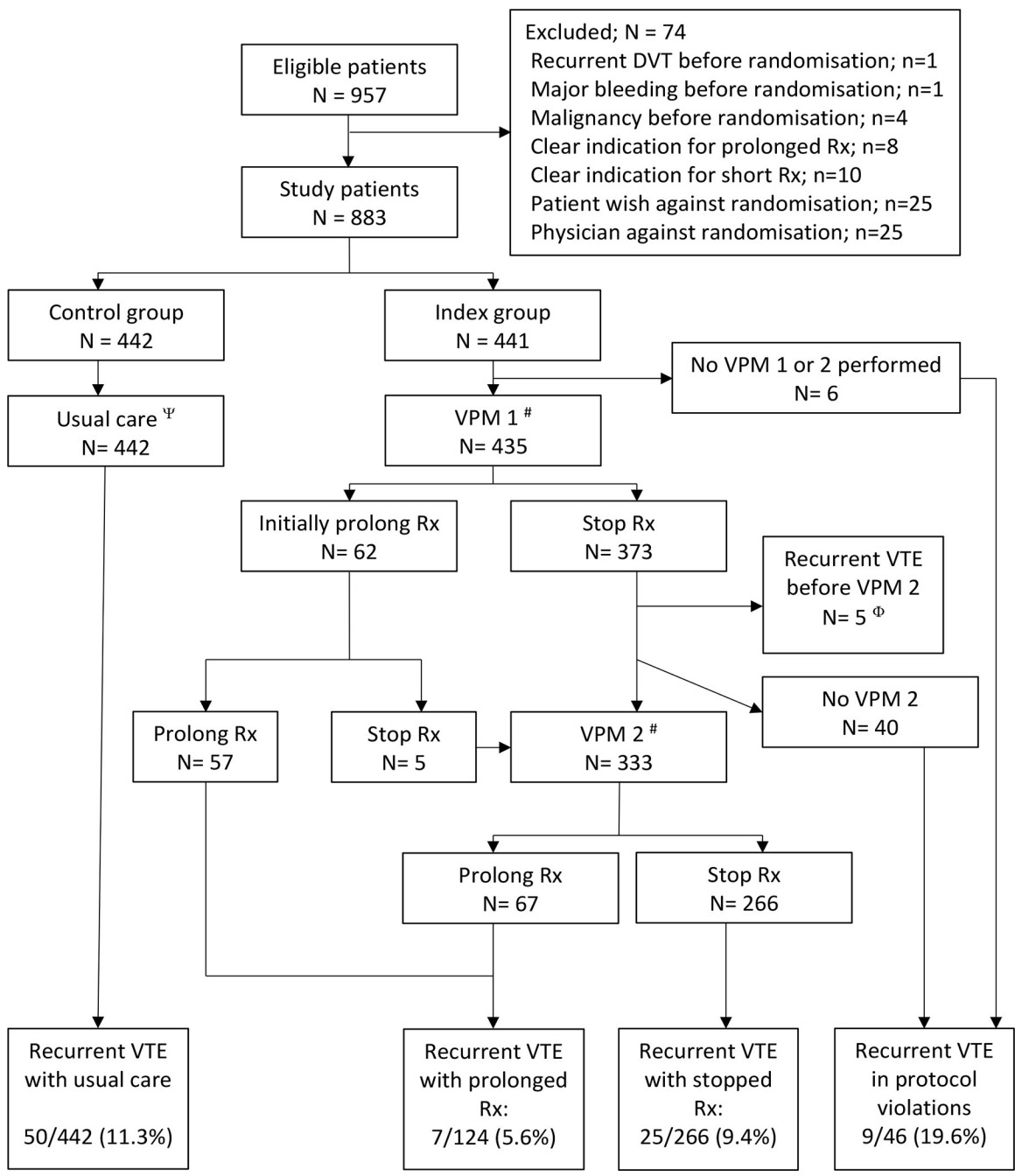

**Fig 1. Flow of patients in the study.** VPM 1 performed shortly before treatment cessation at the end of the initial treatment period; VPM 2 performed approximately 1 month after treatment cessation. $^\Psi$Usual care was left at the discretion of the treating physician; median duration of treatment in this group was 6 months. $^\#$Not all patients followed VPM recommendation: 58 patients were non-adherent (see main text). $^\Phi$These patients had a recurrent VTE event shortly after VPM 1 and thus before VPM 2 (see S2 Table). DVT, deep vein thrombosis; Rx, treatment; VPM, Vienna Prediction Model; VTE, venous thromboembolism.

anticoagulant treatment, not guided by a risk prediction model but rather under usual care at the discretion of the treating physician (see Fig 2).

Rate of adherence to the trial regimen was—as would be expected—not 100%. In fact, 46 patients in the index group did not obtain any risk estimation, and an additional 58 patients

**Table 1. Patient characteristics of randomized patients.**

| Characteristic | VPM patients (*n* = 441) | Control patients (*n* = 442) |
|---|---|---|
| Male sex | 253 (57%) | 254 (58%) |
| Age (years) | 55 (14) | 55 (14) |
| Diabetes mellitus | 24 (5%) | 30 (7%) |
| Previous cardiovascular disease | 26 (6%) | 31 (7%) |
| Index event | | |
| Isolated distal or proximal DVT | 218 (49%) | 216 (49%) |
| PE with or without DVT | 223 (51%) | 226 (51%) |
| Hormonal therapy$ | 90 (20%) | 94 (21%) |
| Known thrombophilia | 44 (10%) | 36 (8%) |
| History of DVT or PE* | 21 (5%) | 21 (5%) |
| Initial treatment duration | | |
| 3 to <6 months | 425 (96%) | 432 (98%) |
| 6 to <12 months | 16 (4%) | 10 (2%) |

Data are presented as number (%) or mean (standard deviation).

$Hormonal therapy included both oral contraceptive use and hormonal replacement therapy, for women only.

*History of DVT or PE more than 10 years ago.

DVT, deep vein thrombosis; PE, pulmonary embolism; VPM, Vienna Prediction Model.

were, for a variety of reasons, unable to comply with the treatment recommendation (i.e., a total of 104 [23.6%]), leading to an adherence rate of 76.4% (Fig 1). These 104 patients were defined as non-adherent in our secondary analysis. Using the 75th quartile of treatment adherence from the propensity model yielded in our secondary analysis a risk ratio similar to that of the primary analysis of 1.02 (95% CI 0.62 to 1.66).

Finally, 5 patients (1%) had a recurrent VTE event shortly after initial treatment was stopped and after VPM 1, and as such did not receive VPM 2 (see S2 Table).

## Predictive performance of the VPM

In the 694 patients from the index and control groups who did not receive prolonged treatment, the VPM showed good discriminative performance, with a c-statistic of 0.76 (95% CI 0.69 to 0.83). However, the model underestimated recurrence risk around and above the threshold of 5% (see Fig 3). For instance, among 134 patients with predicted probability of >4% to 5%, the observed incidence of the outcome was 6.7% (95% CI 2.5% to 10.9%), which we believe will be considered too high by many, although the 95% confidence interval here is also relatively wide due to imprecision. At lower predicted risks, however, the model showed good calibration, for instance exemplified by the observation that in the group with a predicted risk of >2% to 4% (*n* = 284), the observed risk of recurrence was also low, at 2.5% (95% CI 0.7% to 4.3%) (see Fig 4).

## Discussion

### Main findings

The randomized VISTA trial was primarily set up to determine whether applying a risk prediction model to estimate recurrence risk in patients with unprovoked VTE, with subsequent risk-based tailoring of prolonged anticoagulant treatment, would lower the risk of recurrence as compared to usual care. Our findings showed that we were unable to demonstrate such a

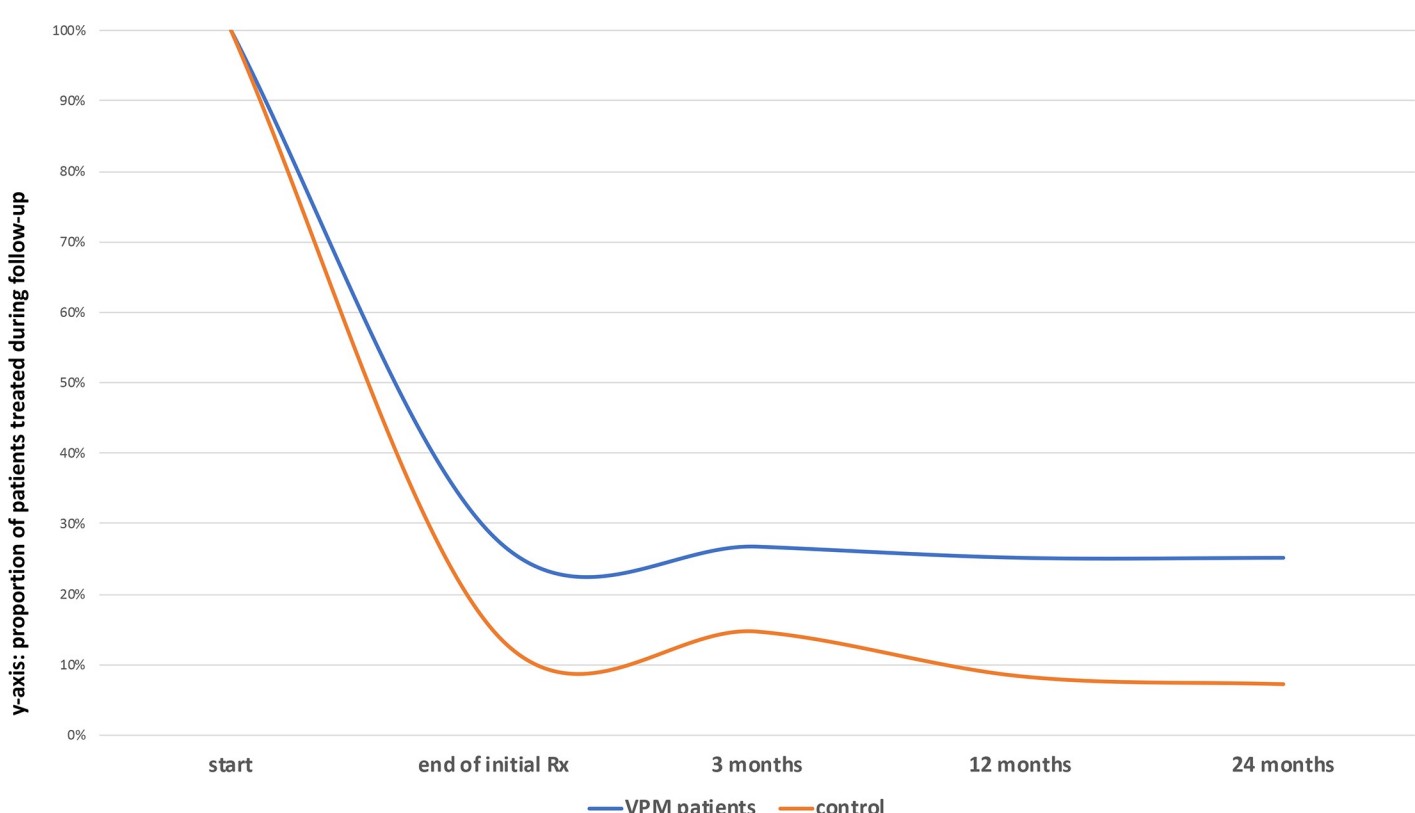

**Fig 2. Patients using anticoagulant treatment during follow-up.** Rx, treatment; VPM, Vienna Prediction Model; VTE, venous thromboembolism.

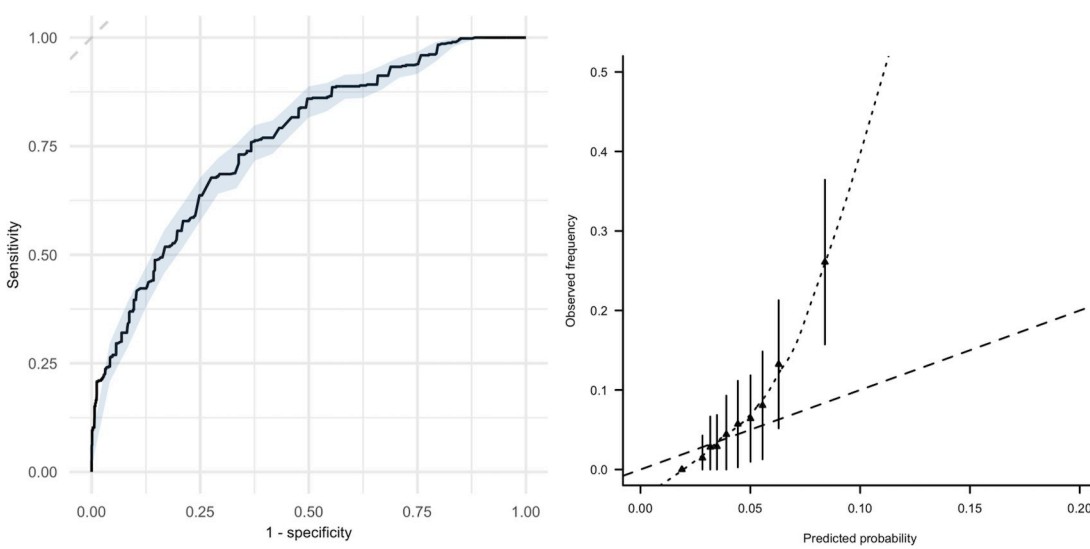

**Fig 3. Discrimination and calibration of the Vienna Prediction Model.** Left panel: discriminative performance of the Vienna Prediction Model with 95% CI (blue shading): c-statistic 0.76 (95% CI 0.74 to 0.78). Right panel: calibration plot comparing predicted risk of recurrent venous thromboembolism by the Vienna Prediction Model (*x*-axis) with observed recurrence rate (*y*-axis).

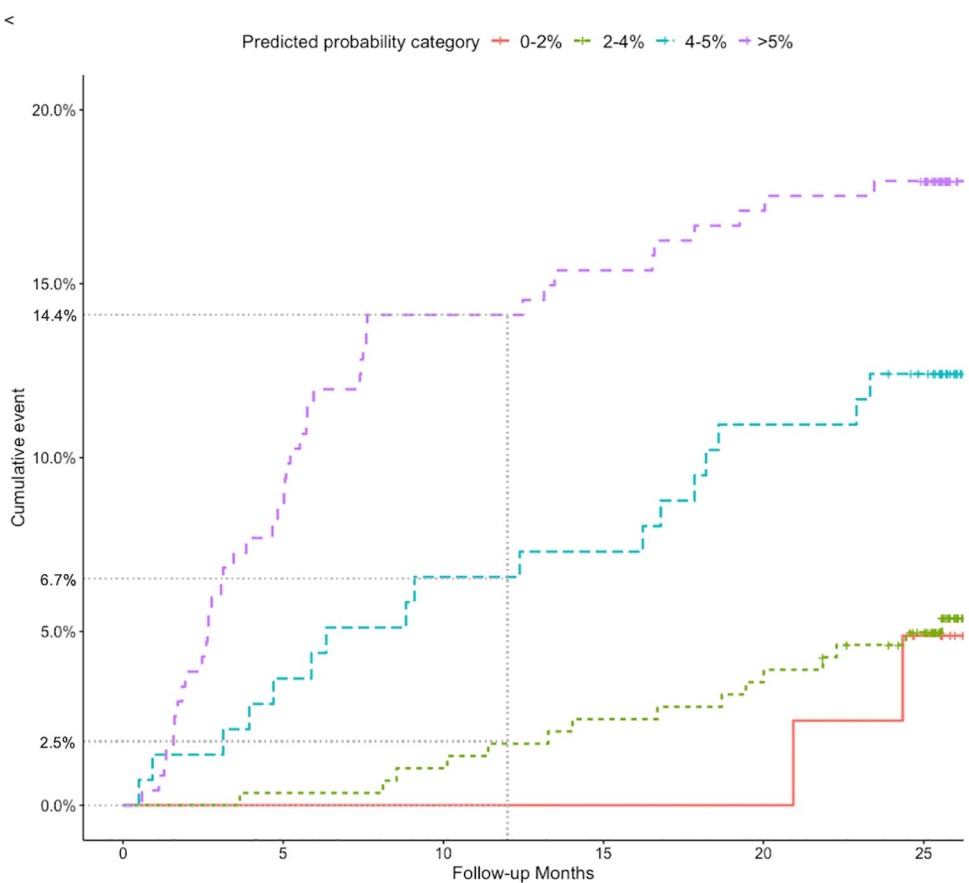

**Fig 4. Cumulative observed VTE risk for different strata of predicted risk.** At 12 months of follow-up, the observed incidence of recurrent VTE was 0% (95% CI 0%–0%), 2.5% (0.7%–4.3%), 6.7% (2.5%–10.9%), and 14.4% (9.9%–18.9%) for the predicted probability categories 0% to 2% (n = 40), >2% to 4% (n = 284), >4% to 5% (n = 134), and higher than 5% (n = 236), respectively. VPM, Vienna Prediction Model; VTE, venous thromboembolism.

reduction. In terms of predicting the risk of recurrent VTE, the VPM that we used showed good discriminative power and moderate to good calibration, in particular at the clinically relevant lower spectrum of predicted risk.

## Comparison with existing literature

Other approaches exist to predict the risk of recurrence in unprovoked VTE, albeit only validated in prospective cohorts, thus without a control group. For instance, Rodger et al. reported on the prospective application of the HERDOO2 rule [12]. This rule classifies all men as high risk (recommending prolonging anticoagulant treatment), and additionally all women with the presence of 2 or more of the HERDOO2 rule items (hyperpigmentation, edema, or redness in either leg; D-dimer level $\geq$ 250 μg/l; obesity [body mass index $\geq$ 30 kg/m$^2$]; or older age, i.e., age $\geq$ 65 years). The study enrolled a total of 2,785 participants, and in total 631 women were classified as low risk (23% of the total study population). In this low-risk group, 591 women actually stopped anticoagulant treatment, and 17 of them had a recurrent VTE event, yielding a recurrence rate of 3.0% per year (95% CI 1.8% to 4.8%). In another study by Kearon et al., the D-dimer test was used to guide the decision of whether or not to extend anticoagulant treatment [13]. In this study, a total of 410 patients were enrolled, and 319 patients had a negative D-dimer test 1 month after treatment discontinuation (i.e., 78% of the total study

population). These patients stopped antithrombotic treatment, and recurrent disease occurred in 42 patients, for a yearly recurrence rate of 6.7% (95% CI 4.8% to 9.0%). Comparing these observational cohorts with non-treated patients in our study, we observed that the VPM showed good discriminative performance for estimating the risk of recurrent VTE. In particular, for patients with lower predicted risks (up to around 4%), the model showed good calibration. At higher predicted risk, the model underestimated recurrence risk in our population, which, importantly, is similar to the results of a pooled individual participant data meta-analysis externally validating the VPM, which also showed that the model tended to underestimate recurrence risk in patients with unprovoked VTE [14]. Yet, should we define low risk as a predicted risk of recurrent VTE up to 4%, the VPM identified 47% (324 out of 694 patients) as low risk. These findings thus compare favorably with the existing observational evidence, with the important observation that the group identified as "low risk" is substantially larger for the VPM as compared to the HERDOO2 rule. Moreover, the observed yearly recurrence rate in this group at a low predicted risk of recurrent VTE is lower when compared to "only" using D-dimer level to guide treatment continuation, as was done in the study by Kearon et al.

## Strengths and limitations

For full appreciation of our findings, several limitations should be discussed. A major strength of our study is the randomized design. The sobering finding that we did not observe a difference in recurrence between a strategy of applying a prediction model versus usual care exemplifies that the impact of prediction models should also be studied in randomized trials.

Several factors may have contributed to the inability of our trial to detect a difference in recurrence. First, the population included in our trial may be considered as being at relatively low risk of recurrence. For instance, the observed risk of recurrence was only slightly higher than 5% per year in the control group, albeit this may partly be explained by the fact that, in at least part of the control group, treatment was prolonged. Additionally, we included women with oral-contraceptive-related VTE, which could be considered debatable given that oral contraceptive use is currently seen as a minor transient risk factor in the most recent definitions from ISTH [15].

Second, given the pragmatic nature of our trial, not all patients complied with the treatment recommendation of the VPM. Similarly, some hospital specialists caring for patients in our control group may have been influenced due to the trial participation of others of their patients. Nevertheless, they did not receive any information on the result of D-dimer testing. Yet obviously, both issues may dilute any effect that would be observed should all patients and physicians have been "forced" to be compliant with our risk-tailored approach. Therefore, we performed secondary (post hoc) analyses, which yielded similar results. For these secondary analyses, we used a propensity-model-based approach, comparing patients (between the index and control arms) above the 75th propensity quartile of being adherent. Albeit clinically an intuitive approach, other more advanced statistical approaches exist, such as inverse probability weighting [16]. Prompted by *PLOS Medicine* reviewer comments, we used these approaches as well, which yielded very similar inferences (see S2 Text for these additional analyses).

Third, during study inclusion, patient accrual rates in our trial slowly dropped, likely due to the introduction of NOAC treatment (which is not managed by thrombosis services, thus these patients were not identified by our study-protocol-defined recruitment approach). This changing anticoagulant landscape that occurred exactly during the conduct of the VISTA trial, as well as budget constraints, prevented us from changing our study design or prolonging patient accrual into our study. We believe our observations permit us to draw conclusions on the effect size of implementing the VPM on recurrence risk in patients with unprovoked VTE,

given that this effect size is at the line of unity. Moreover, our study sample did allow us to a draw firm conclusion on the predictive performance of the VPM.

Fourth, recurrent VTE evens were classified after scrutinizing all available hospital discharge information and as such were adjudicated in a non-blinded manner. Similarly, this was a pragmatic trial, and we did not use blinded treatment. However, we believe the potential influence of these biases is likely very low (or even negligible) in our trial. Hospital specialists were not actively involved in this trial as participating study sites, nor was there any motive for them to favor the outcomes in the intervention or control arm of our trial.

Finally, our analyses of the predictive performance of the VPM were restricted to those not receiving anticoagulant treatment (*n* = 694), thus to relatively lower risk patients (as higher risk patients were more likely to receive prolonged anticoagulant treatment). Therefore, prompted by *PLOS Medicine* reviewer comments and to explore the potential influence of this analysis approach on the predictive performance of the VPM, we performed 2 types of additional analyses: (i) including both treated and non-treated patients and (ii) using inverse probability weighting to account for the treatment effects of anticoagulant treatment [17]. These analyses yielded similar inferences on the performance of the VPM (see S2 Text for detailed explanation of these analyses).

## Clinical applicability and research implications

Given our inability to demonstrate a reduction in recurrence risk by risk-stratifying care, and in light of the publication of recent extended NOAC trials that showed superiority of extended NOAC treatment over either placebo or aspirin for reducing recurrence risk in patients with (mainly) unprovoked VTE, it may be tempting to consider prolonging anticoagulant treatment in all patients with unprovoked VTE [18,19]. Nevertheless, in these trials, follow-up duration was still limited and frail elderly were underrepresented, as were patients with a high bleeding risk, all limiting generalizability to real-life clinical care. Moreover, our findings do indicate that in a large proportion of patients (nearly half of the population included in our trial), a low predicted risk (up to an annualized rate of 4%) is correlated with a similarly low observed risk, suggesting that it might be safe to stop anticoagulant treatment. The threshold for what then defines "low VTE recurrence risk," or what residual recurrence risk is still acceptable, is open for clinical debate. Prompted by *PLOS Medicine* reviewer comments, we additionally explored the following hypothetical cohort of patients: Assume a hypothetical cohort of 1,000 patients with unprovoked VTE. Based upon VISTA observations, 950, 600, and 430 patients will be classified as having an annualized recurrence risk >2% to 4% (low risk), >4% to 5% (moderate risk), or >5% (high risk). Using the observed rates of recurrence from VISTA for these strata, as well as the known hazard ratio of 0.25 for prolonged anticoagulant treatment (compared to placebo [18,19]), we can estimate the number of recurrent VTE events for different treatment strategies over a time period of 5 years. Similarly, this can be done for the number of bleeding events, assuming 3 different major bleeding rates (0.5%/year, 1.0%/year, and 1.5%/year). See S3 Table and S2 Text for further and detailed explanation. Next, we can plot the number of additional VTE and major bleeding events induced by prolonging anticoagulant treatment for certain risk thresholds above which patients are recommended to prolong treatment. In Fig 5, it may be appreciated that this threshold should perhaps be somewhere around 3% to 4%, as then the number of additional VTE events induced because we treat fewer patients is more or less the same as the number of additional bleeding events that occur given that antithrombotic treatment strategy. This threshold is lower than 5%, which is currently still the recommended threshold of "low risk" as defined by ISTH standards. Also, we should acknowledge that in particular at this threshold of 5%, the

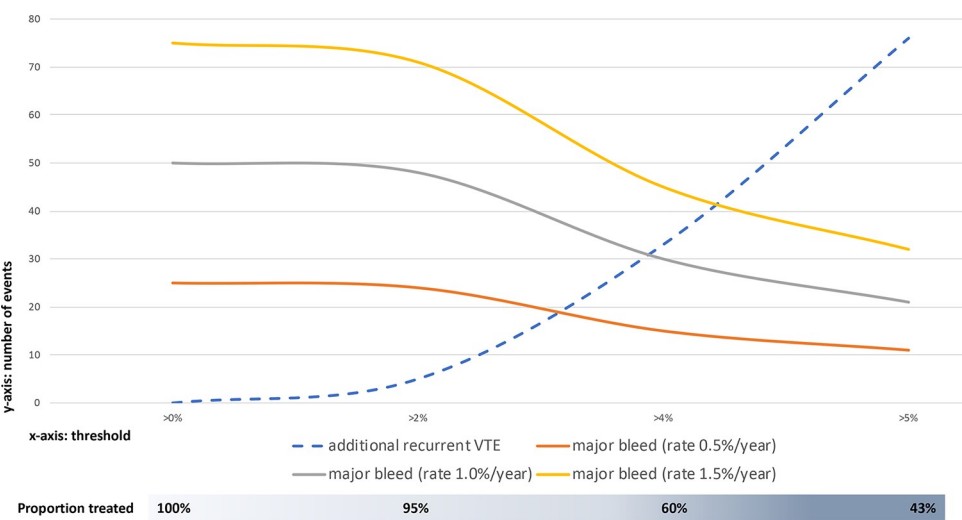

**Fig 5. Clinical tradeoff between additional VTE and major bleeding in a hypothetical cohort in 1,000 VTE patients.** The *x*-axis gives the threshold above which prolonged anticoagulant treatment is recommended, based upon annualized risk estimates from the Vienna Prediction Model. The *y*-axis gives the number of additional VTE and major bleeding events induced depending on the threshold above which prolonged anticoagulant treatment is recommended. The bar below the graph illustrates the proportion of patients treated with anticoagulants depending on the threshold above which prolonged anticoagulant treatment is prescribed. Major bleeding rates are as expected for prolonged non-vitamin K antagonist oral anticoagulant treatment, based upon available trial evidence. VTE, venous thromboembolism.

VPM is less well calibrated, further exemplifying that perhaps a threshold of 3% or 4% is more likely to be clinically meaningful.

## Conclusions

Our results show that an assistive risk-based treatment strategy in all patients with unprovoked VTE is unlikely to reduce recurrence risk as compared to usual care. In those with a low predicted risk as estimated by the VPM, the observed recurrence rate is similarly low, suggesting that it might be safe to stop anticoagulant treatment in these patients.

## Supporting information

**S1 CONSORT Checklist. CONSORT checklist for the VISTA trial.**
(DOC)

**S1 Table. Comparison of patients included in the ITT analysis and patients with non-adherence as included in the PP analysis.** DVT, deep vein thrombosis; ITT, intention to treat; PE, pulmonary embolism; PP, per protocol.
(DOCX)

**S2 Table. Description of 5 patients with a recurrent venous thromboembolic event in the interval between VPM 1 and VPM 2. All DVT index events were proximal.** [$]D-dimer concentration as obtained at the first time the VPM was obtained (VPM1). [#]Depicts the days after VPM 1 is performed (see Fig 1). DVT, deep venous thrombosis; F, female; M, male; PE, pulmonary embolism.
(DOCX)

**S3 Table. Clinical outcomes in a hypothetical cohort of 1,000 VTE patients followed up for 5 years.** [#]These estimates come from the following line of reasoning: We know that the proportion of patients categorized in the following risk strata of annualized VTE recurrence risk of 0% to 2%, >2% to 4%, >4% to 5%, and >5% are 4.9%, 35.1%, 17.1%, and 42.9%, thus 49, 351, 171, and 429 patients in a hypothetical cohort of 1,000 patients, respectively. Based upon our VISTA data, we also know the observed annualized rates of recurrent VTE in these risk strata if anticoagulant treatment is withheld, i.e., 0.0%, 2.5%, 6.7%, and 14.4%, respectively. We know from the NOAC extension trials that these risks of recurrent VTE can be lowered with NOAC treatment, with an observed hazard ratio in these trials of about 0.25, thus yielding annualized rates of recurrent VTE, if managed with prolonged NOAC treatment, of 0.0%, 0.63%, 1.68%, and 3.6%, respectively. Now, let's assume the annualized rates of recurrent VTE to be constant over a 5-year period. This is a conservative assumption, as we know that the rates of recurrent VTE are highest in the first year after the initial event and typically a little bit lower in the subsequent years. Using these data, we can estimate the number of VTE events over a 5-year period in our hypothetical cohort of 1,000 VTE patients for the following scenarios: (i) prolong treatment in all patients, (ii) prolong only if the estimated risk of recurrence is >2%, (iii) prolong only if the estimated risk of recurrence is >4%, (iv) prolong only if the estimated risk of recurrence is >5%, or (v) do not prolong anticoagulant treatment in any patients. In fact, the number of recurrent VTE events for these scenarios would be 103, 103, 135, 178, and 410 events, or, compared to prolonging treatment in all patients (first scenario), 0, 0, 33, 76, and 308 additional events, respectively. Similarly, we can estimate the number of major bleeding events induced in these different scenarios. Unfortunately, the annualized rate of major bleeding in VTE patients on prolonged NOAC treatment is less certain, so let's assume the number of major bleeding events for 3 different annualized rates of major bleeding: 0.5%/year, 1.0%/year, and 1.5%/year. Over a 5-year period, we can then calculate the number of major bleeding events for these different rates of major bleeding. (DOCX)

**S1 Text. Protocol of the VISTA trial as approved by the ethics committee.** (PDF)

**S2 Text. Detailed explanation of additional analyses prompted by reviewers from *PLOS Medicine*.** (DOCX)

# Acknowledgments

The authors would like to thank Anna E. C. Kingma for her help in data collection, as well as all patients and physicians, and the participating thrombosis services for their collaboration while conducting this trial.

# Author Contributions

**Conceptualization:** Ruud Oudega, Roger E. G. Schutgens, Karel G. M. Moons.

**Data curation:** Geert-Jan Geersing.

**Formal analysis:** Geert-Jan Geersing, Nicolaas P. A. Zuithoff, Kit C. Roes, Toshihiko Takada.

**Funding acquisition:** Geert-Jan Geersing, Ruud Oudega, Roger E. G. Schutgens, Karel G. M. Moons.

**Investigation:** Geert-Jan Geersing, Janneke M. T. Hendriksen.

**Methodology:** Geert-Jan Geersing, Kit C. Roes, Toshihiko Takada, Karel G. M. Moons.

**Project administration:** Geert-Jan Geersing, Janneke M. T. Hendriksen.

**Supervision:** Geert-Jan Geersing, Nicolaas P. A. Zuithoff, Kit C. Roes.

**Validation:** Geert-Jan Geersing.

**Writing – original draft:** Geert-Jan Geersing.

**Writing – review & editing:** Janneke M. T. Hendriksen, Nicolaas P. A. Zuithoff, Kit C. Roes, Ruud Oudega, Toshihiko Takada, Roger E. G. Schutgens, Karel G. M. Moons.

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
