## [Decision Letter · Decision Letter 0]

13 Feb 2020

Dear Dr. Geersing,

Thank you very much for submitting your manuscript "Tailoring of anticoagulant treatment by a recurrence risk prediction model in patients with venous thrombo-embolism compared to usual care; a validation and impact study." (PMEDICINE-D-20-00044) for consideration at PLOS Medicine. 

[LINK]

In light of these reviews, I am afraid that we will not be able to accept the manuscript for publication in the journal in its current form, but we would like to consider a revised version that addresses the reviewers' and editors' comments. Obviously we cannot make any decision about publication until we have seen the revised manuscript and your response, and we plan to seek re-review by one or more of the reviewers. 

We expect to receive your revised manuscript by Feb 28 2020 11:59PM. Please email us (plosmedicine@plos.org) if you have any questions or concerns.

We look forward to receiving your revised manuscript. 

Sincerely,

Adya Misra, PhD

Senior Editor 

PLOS Medicine

plosmedicine.org

Article meta-data- please ensure you provide article meta-data such as financial information, competing interests and data availability statements. If you have questions about this please contact us at plosmedicine@plos.org

Title- Please revise your title according to PLOS Medicine's style. Your title must be nondeclarative and not a question. It should begin with main concept if possible. "Effect of" should be used only if causality can be inferred, i.e., for an RCT. Please place the study design ("A randomized controlled trial," "A retrospective study," "A modelling study," etc.) in the subtitle (ie, after a colon).

Please structure your abstract using the PLOS Medicine headings (Background, Methods and Findings, Conclusions).

Abstract- could you include a sentence to describe unprovoked venous thrombo-embolism

Please complete the CONSORT checklist and ensure that all components of CONSORT are present in the manuscript, including [how randomization was performed, allocation concealment, blinding of intervention, definition of lost to follow-up, power statement]. 

Abstract Methods and Findings: 

* Please ensure that all numbers presented in the abstract are present and identical to numbers presented in the main manuscript text. 

* Please include the study design, population and setting, number of participants, years during which the study took place, length of follow up, and main outcome measures.

 * Please quantify the main results (with 95% CIs and p values)

* Please include a summary of adverse events if these were assessed in the study. * In the last sentence of the Abstract Methods and Findings section, please describe the main limitation(s) of the study's methodology.

Abstract primary outcome does not match the primary outcome noted in the CT registry, please clarify 

Please provide brief information about the VISTA study and introduce the term on first view 

Line 102, study protocol URL appears to be the journal URL, please provide the study protocol as SI files or provide the correct link 

The information about ethics approval and written informed consent should ideally be provided together within the methods section

Please provide dates of patient recruitment and follow up more clearly in the methods section. 

The CONSORT checklist should be cited within the methods section and page numbers should be removed as these are likely to change during publication. Please instead refer to paragraphs and sections. 

Comments from the reviewers:

Reviewer #1: This article presents the results of a randomised trial testing whether using the Vienna prediction model decreases the incidence of recurrent venous thrombo-embolism (VTE) compared to usual care in patients with previous unprovoked VTE. It also seeks to assess the predictive ability of the Vienna model by treating the study as a cohort. While the results are clearly presented and I have no major concern, I have a number of minor comments listed below:

Major comments:

* No major comments.

Minor comments:

* Please update the title to include the study design as per PLOS Medicine conventions.

* In the conclusion of the abstract, please consider changing the last sentence to make the statement less definitive/certain. For example, by writing: "in those with a low predicted risk of recurrence, the observed rate was also low; suggesting that it might be safe to stop anticoagulant treatment in these patients".

* Please clarify whether the primary outcome (see lines 167-168) was measured 24 months after randomisation or 24 months after the end of this initial treatment phase. I believe this is in fact the same time point but it would be good to make it more explicit when defining the outcome.

* Please clarify whether outcome assessment was performed and/or adjudicated in a blinded manner and, if not, please discuss the potential risks associated with reporting/ascertainment bias.

* Line 221 mentions imputed datasets; however, data imputation is not discussed prior to this statement (it is mentioned in the last sentence of the methods). Please clarify the nature of the imputation including variables imputed, variables included in the imputation model and the number of imputed datasets. Please also indicate whether the imputation was pre-specified in the protocol or analysis plan or post-hoc.

* When restricting the analysis to patients with a high estimated probability of adherence, please report the degree of comparability between the index and control group. i.e. were they well matched or did some baseline imbalance remain between the two groups after selecting those with a high probability of adherence? Please consider adding the corresponding baseline table to the online supplement. In case of baseline imbalances (e.g. standardised difference greater than 0.1), please consider adjusting the analysis.

* I am confused by the fact that the predictive performance analysis excluded patients who continued anticoagulant treatment. In patients randomised to the index group, only those at higher risk (according to the Vienna prediction model itself) would have continued anticoagulant treatment. Therefore, excluding these patients limits the validation exercise to only those at lower risk. Please confirm/clarify.

* Please clarify why 12 months of follow-up were selected for the validation when the primary outcome of the trial was measured at 24 months.

* The authors (lines 231-232) mention using the baseline hazard function from the original publication. Does this imply that the risk score used in the validation is different to the risk score used in the randomised trial? If so, please clarify the reason for choosing different equations.

* Figure 2: It is difficult to see the proportions in white font over the bubbles even after downloading the original .tiff version. Please consider using a different font and/or color.

* The 'validation' analysis suggest that the authors were able to calculate the risk from the Vienna prediction model in both the index and control group. Please consider reporting the number and proportion at high risk separately for each randomised group. This could potentially be added to Table 1.

* I would suggest moving Table 2 (Description of 5 patients with interval recurrent venous thrombo-embolic event) to the online supplement as it does not appear essential to the main 'story'.

* The authors state that "the model underestimated recurrence risk around and above the threshold of 5%" (see lines 318-319); however, according to Figure 3 (right panel) the predicted line seems very close to every observed estimate (triangles) and certainly well within the confidence bands. Please clarify.

* Pending clarification of the above comment, could it be that the 'apparent' underestimation around the 5% threshold is linked to the fact that some (most?) patients with a risk above 5% continued treatment and were therefore excluded from the validation exercise. To address this concern, I would encourage the authors to consider repeating the validation analysis without excluding any patient (i.e. regardless of whether they continued treatment) as a sensitivity analysis.

* I believe that the data should be made publicly accessible instead of available on request as per PLOS Medicine policy but will defer to the editor's guidance on this aspect.

-Laurent Billot

Reviewer #2: The manuscript by Geersing et al. describes the results from a randomized trial conducted with the aim of comparing decision making of treatment duration of oral anticoagulant treatment after venous thromboembolism (VTE) by use of the VIENNA prediction model (VPM). The overall hypothesis was lower recurrence risk among patients assessed using a risk-based strategy (VPM) than recurrence risk found in a control group. In the VPM arm, a total of 441 was enrolled and 442 VTE patients were randomized to usual care. The duration of follow-up was 24 months.

The study was an investigator-initiator trial conducted on multiple sites with block-randomizing (site and type of VTE) of patients to either VPM or usual care. The trial followed principles of a pragmatic trial with no blinding. A CONSORT description was available and the trial was registered in a Dutch trial database. Participants presented with VTE and were eligible if no so-called 'provoking' risk factors were prevalent including active cancer. Indications for long-term treatment with an oral anticoagulant treatment was also an exclusion criteria.

Initially all patients were treated with warfarin 3 to 6 months after low molecular weight heparin lead-in. Before end of this treatment period, patients were randomized to either usual care (discretion of the physician) or to a risk-tailored approach using VPM.. Extended (i.e. longer duration of the initial three to six months) duration of treatment was offered to those with a predicted risk of 5% per year or above in accordance with the ISTH recommendations. Model risk assessment was done twice, first time short before treatment cessation and second time approximately one month after. In the control group, the treatment cessation time was based on the discretion of the physician. Patients were followed up at 3, 12, and 24 months.

Major comments:

The current clinical equipoise relates to when oral anticoagulant treatment can safely be withheld in patients presenting with VTE and no apparent clinical risk factors. Anticoagulation is effective in preventing VTE recurrence but the use of oral anticoagulant treatment has an inheriting risk of bleeding. Hence, the duration of treatment should be based on an assessment of the net clinical benefit. Therefore, it is important to present the proportion of patients (stratified according to randomization) with treatment duration in e.g. quartiles, which has been the traditional length of treatment in VTE patients (3, 6, 9, or 12 months). This would allow the reader some insights into the initial decision making that was left to the discretion of the treating physician. Preferably, this could also be sub-categorized into patients presenting with DVT or PE.

It is not entirely clear when the randomization occurred. Since this was a pragmatic (non-blinded) study design, it could be hypothesized that knowing the assignment of the individual treatment arm would affect the behavior (and preferences) of the patients and/or physicians. The authors are encouraged to reflect on the consequences of using a no-blinded study design, and on the direction and magnitude of this potential bias on the effect estimates. Furthermore, the Vienna model consists of three parameters: the extent of VTE, patient sex, and D-dimer concentration. This is obvious information, also for physicians determining anticoagulant treatment duration without VPM. In consequence, this is likely to result in no difference between the two treatment arms. This dilemma could be discussed in the article. 

The secondary analysis, which was not pre-planned, is of interest as these estimates may reflect a more realistic effect of what would be seen in everyday clinical practice. However, the implementation of the approach is lacking details. The authors are encouraged to consult this reference DOI: 10.1056/NEJMsm1605385, which outlines an appropriate analytic plan to conduct per-protocol analyses specifically for pragmatic trials that minimizes the risk of selection bias in relation to censoring. Additionally, the following sentence is unclear: (p7) "we used patients within the 75th quartile of predicted probabilities of protocol adherence." Was the data restricted based on the propensity score? If so, it would be important to characterize the patients who did not enter this analysis - what were the main differences in terms of baseline characteristics? Perhaps an approach like inverse probability of treatment (or rather protocol adherence) weighting could allow for a more efficient use of the available data?

What is the argument for assessing performance only among patients who did not receive prolonged treatment? This would be a selected group of patients with a presumed low recurrence risk. Calibration in high risk patients would be of clinical relevance. 

The performance, in terms of calibration, of VPM is assessed using 4 probability categories (0-2%; 2-4%; 4-5%; and >5%). However, the clinical relevant cut off point is at 5%, as stated by the International Society of Thrombosis and Hemostasis. At this threshold the VPM is not well calibrated. The authors should deliberate on the consequence of bad calibration at this clinically meaningful cut-off point.

It would be relevant to include relative estimates with 95% CI between treatment arms on risk of bleeding.

- 

Minor comments:

Could the authors please check the order of the figures/tables in the supplemental material?

Please provide labels on the y-axis for Figure 2.

What is meant with the term "impact study" in the title? Consider to leave it out.

The study excludes patients with a history of VTE within 10 years. Yet, table 1 list patients with a "history of DVT or PE". Please clarify, if this is a VTE more than 10 years ago? (and supplementary table S1)

Table 1: proportion of patients receiving hormonal therapy - please clarify if this proportion is for women only. (and supplementary table S1)

It could be mentioned (and discussed) that VPM has previously been externally validated. 

It could be highlighted that the authors are not identical with the ones developing the VPM. 

Table 1: "Index" could be rephrased to "VPM patients"

Figure 1: 10 patients are excluded because of clear indication for short treatment. Why and what is this?

Reviewer #3: In this randomized controlled trial, the Vienna Prediction Model (VPM) was tested to determine whether implementing the model would reduce recurrence risk. 883 patients were randomised to using the VPM compared with usual care. No difference in number of recurrent VTE events was observed, and slightly more CRNM bleeding in the intervention arm. The authors conclude that application of the VPM in patients with unprovoked VTE is unlikely to reduce overall recurrence risk.

This is a well-described, honest and important account of how a promising prediction model can fail to translate in improved clinical care, and therefore imperative to be noticed by both clinicians and researchers. For this reason it is also, possibly even more so than for 'positive' studies, necessary to describe in detail what happened.

Most important comments:

1) This study in fact investigates 2 things at the same time: the performance of the VPM and the related antithrombotic strategy. The authors now conclude that the absence of a difference in risk is related to the application of the model. However, it could also simply be due to insufficient treatment related to the result of the model. Theoretically the recurrence risk in both study arms should be equal. So, to find a difference in actual VTE occurrence, a strategy should have been in place that would strongly impact this risk. This has not really happened, so it seems, considering that the rate of prolonged treatment was in fact low in both arms, although a bit higher in the intervention arm: about 25 vs 10% (if I infer correctly from fig 2). So all effect from applying the model should have come from this small difference in treated subjects, which seems implausible. This option, of an insufficient treatment difference between the arms, could receive more attention in the Discussion.

2) In fact I think it could be modelled (e.g. as in a Mendelian Randomisation analysis) how many more cases could have been prevented if the difference should have been larger between the groups (and likewise the bleedings). This would clearly not be the same as in a true RCT but it might shed some light on the respective role of the VPM and the applied treatment strategy in the observed results of this study. 

3) Why was the rate of treatment so low anyway? Was only such a small proportion of patients classified above 5% per year? I find this information not easy to find in the manuscript: which patients took OAC or not at what predicted risks and what were their observed risks, with and without treatment. This would also give some information on effectiveness of the treatment given equal estimated risks. It could also be discussed how the low treatment rate relates to the current guidelines in which everybody with an unprovoked first event is recommended to continue unless a high bleeding risk is present. 

Other comments:

4) Please mention how the criteria used for defining an unprovoked event relate to other definitions (from other scores or guidelines, such as the ISTH's)

5) " Risk estimation was performed twice, namely shortly before treatment cessation at the end of the initial treatment period (thus while still using anticoagulant treatment) and approximately one month after treatment cessation" This is rather confusing for both patients and doctors? Could this have affected the results and would this strategy impede implementation of the model?

6) How were possible recurrent events adjudicated as such?

7) Were all patients followed for 24 months or until recurrence occurred? Not clear if people were lost to follow-up? 

8) " 46 patients in the index group did not obtain any risk estimation whereas an additional 58 patients (total of 104) were for a variety of reasons unable to comply to the treatment recommendation" It is unclear which 58 patients is referred to from fig 1.

[LINK]

---

## [Decision Letter · Decision Letter 1]

21 Apr 2020

Dear Dr. Geersing,

Thank you very much for re-submitting your manuscript "Effect of tailoring anticoagulant treatment duration by applying a recurrence risk prediction model in patients with venous thrombo-embolism compared to usual care: a randomized controlled trial." (PMEDICINE-D-20-00044R1) for review by PLOS Medicine.

I have discussed the paper with my colleagues and the academic editor and it was also seen again by xxx reviewers. I am pleased to say that provided the remaining editorial and production issues are dealt with we are planning to accept the paper for publication in the journal.

[LINK]

We look forward to receiving the revised manuscript by Apr 28 2020 11:59PM. 

Sincerely,

Adya Misra, PhD

Senior Editor 

PLOS Medicine

plosmedicine.org

Requests from Editors:

Abstract

Please mention the aim in the background section

Please state where the trials took place and provide brief patient demographics 

Conclusions should start with “our results show” or similar 

On page 3 please include the sub heading “Author Summary” and include bullet points for readability

Please note all references should be within square brackets and the bibliography should be modified to Vancouver style

The Data Availability Statement (DAS) requires revision. For each data source used in your study: a) If the data are freely or publicly available, note this and state the location of the data: within the paper, in Supporting Information files, or in a public repository (include the DOI or accession number). b) If the data are owned by a third party but freely available upon request, please note this and state the owner of the data set and contact information for data requests (web or email address). Note that a study author cannot be the contact person for the data. c) If the data are not freely available, please describe briefly the ethical, legal, or contractual restriction that prevents you from sharing it. Please also include an appropriate contact (web or email address) for inquiries (again, this cannot be a study author).

Methods

Line 121- could you provide the name of the local thrombosis services or mention the names of cities where these services are based (as you have in line 132)

Please remove the reference to the study protocol online and add a brief sentence that this information is in the SI files 

Please correct November 30rd to 30th 

Please provide citations to or copies of questionnaires used in the study as Si files

Please move the role of the funding source to the financial disclosure section of the article meta-data

Line 453- we do not allow instances of data not shown, so please provide these data or remove this sentence

Please revise the discussion for readability, as it is currently rather long and the main messages are somewhat lost. The discussion of strengths, limitations, clinical application of your work is important but please do see if these sections can be shortened. For instance: Lines 486-496 can be considered for removal. I appreciate the additional information regarding the hypothetical cohort here and the discussion around the 5% low risk threshold, but I would recommend shortening this section so that the main message is clearly highlighted for the reader. 

Figure 5-can this be placed within supplementary information?

Conclusions

Please start this section with “our results show” or something similar

This section requires some revision, as it currently doesn’t seem to sum up you work sufficiently clearly. Could you rephrase Lines 537-539 so clarify your findings about low risk recurrence 

Fig1 – could you ensure this adheres to CONSORT style

Fig 2- the X-axis needs a label

Comments from Reviewers:

Reviewer #1: The authors have responded thoroughly to my queries and conducted a range of useful sensitivity analyses. 

I note that in a few instances, the results of the sensitivity analyses are not reported. Instead a sentence mentions that the results were consistent with the main analyses and adds "(data not shown)". I would encourage the authors to include the results of the sensitivity analyses in the supplement or, in the case of a short result such as a point estimate and its confidence interval, to include it in the text.

Please try and make as explicit as possible which analyses were pre-planned vs post-hoc (e.g. multiple imputations). I would also encourage the authors to explicitly state which analyses were performed to address comments from the PLOS Medicine reviewers.

-Laurent Billot

Reviewer #2: The authors have responded sufficient to the comments raised.

Reviewer #3: all my comments were well addressed

[LINK]

---

## [Editor Report · Decision Letter 2]

3 Jun 2020

Dear Dr. Geersing, 

On behalf of my colleagues and the academic editor, Dr. Suzanne C. Cannegieter, I am delighted to inform you that your manuscript entitled "Effect of tailoring anticoagulant treatment duration by applying a recurrence risk prediction model in patients with venous thrombo-embolism compared to usual care: a randomized controlled trial." (PMEDICINE-D-20-00044R2) has been accepted for publication in PLOS Medicine. 

PRODUCTION PROCESS

PRESS

PROFILE INFORMATION

Thank you again for submitting the manuscript to PLOS Medicine. We look forward to publishing it. 

Best wishes, 

Adya Misra, PhD

Senior Editor 

PLOS Medicine

plosmedicine.org